# Skin color-specific and spectrally-selective naked-eye dosimetry of UVA, B and C radiations

Wenyue Zou[1], Ana González[2], Deshetti Jampaiah[1], Rajesh Ramanathan [1], Mohammad Taha[3], Sumeet Walia[3], Sharath Sriram [3], Madhu Bhaskaran[3], José M. Dominguez-Vera [2] & Vipul Bansal [1]

Spectrally–selective monitoring of ultraviolet radiations (UVR) is of paramount importance across diverse fields, including effective monitoring of excessive solar exposure. Current UV sensors cannot differentiate between UVA, B, and C, each of which has a remarkably different impact on human health. Here we show spectrally selective colorimetric monitoring of UVR by developing a photoelectrochromic ink that consists of a multi-redox polyoxometalate and an e$^-$ donor. We combine this ink with simple components such as filter paper and transparency sheets to fabricate low-cost sensors that provide naked-eye monitoring of UVR, even at low doses typically encountered during solar exposure. Importantly, the diverse UV tolerance of different skin colors demands personalized sensors. In this spirit, we demonstrate the customized design of robust real-time solar UV dosimeters to meet the specific need of different skin phototypes. These spectrally–selective UV sensors offer remarkable potential in managing the impact of UVR in our day-to-day life.

[1] Ian Potter NanoBioSensing Facility, NanoBiotechnology Research Laboratory, School of Science, RMIT University, Melbourne, VIC 3000, Australia. [2] Departamento de Química Inorgánica and Instituto de Biotecnología, Universidad de Granada, Granada 18071, Spain. [3] Functional Materials and Microsystems Research Group and Micro Nano Research Facility, RMIT University, Melbourne, VIC 3000, Australia. Correspondence and requests for materials should be addressed to J.M.D-V. (email: josema@ugr.es) or to V.B. (email: vipul.bansal@rmit.edu.au)

Skin cancer, one of the most common cancers globally, is mainly caused by an excessive exposure to ultraviolet radiation (UVR)[1]. UVR is classified into UVA (315–400 nm), UVB (280–315 nm), and UVC (100–280 nm). The longwave UVA penetrates deep into the skin and its cumulative exposure over lengthy periods results in skin aging and wrinkling[2]. UVB is particularly effective at damaging DNA and short bursts of high UVB doses cause sunburn, increasing the likelihood of developing skin cancer and cataract[3]. The shortwave UVC has the highest energy that can induce deadly damages to all lifeforms, including humans, plants, and planktons[4,5]. Fortunately, the atmosphere filters out most of the solar UVB and all UVC. Therefore, the solar UVR that reaches the earth surface is a combination of UVA and UVB. Further, due to the depletion of the stratospheric ozone layer, our natural protective filter is progressively declining. This is leading to higher levels of UVR reaching the earth surface[6,7]. Particularly, in regions such as Australia, where the ozone layer is significantly depleted, it is estimated that at least 2 in 3 people will be diagnosed with skin cancer before the age of 70[8].

UVR also offers certain benefits and has found applications in the prevention or treatment of rickets, lupus vulgaris, and psoriasis[9–11]. A modest dose of UVB is essential for the body as it interacts with 7-dehydrocholesterol in the skin to produce Vitamin D[12]. However, whilst UVB stimulates the production of Vitamin D, UVA may destroy it[13], thereby keeping the body in balance. Considering that a balanced dose of sun exposure is critical to maintaining body's functionality, and yet the therapeutic benefits of solar rays cannot outpace the associated ramifications; an efficient solar UV monitoring sensor is of paramount importance for safe and effective sun exposure. Further, as UVC is utilized in a number of industrial processes, the monitoring of UVC remains equally critical[14,15]. In particular, the ability of UVC to kill bacteria, virus, yeast, and fungi has seen a rapid rise in the use of UVC germicidal lamps in homes and hospitals for water and air disinfection, and in food processing industries to extend the shelf-life of products[14]. Even the smallest levels of UVC exposure without appropriate protection can induce serious damage[16,17].

UVA, B, and C not only cause remarkably different effects on biological entities and man-made products; the unpredictable composition of UVR on the earth surface adds another level of complexity in their selective monitoring[18,19]. Factors influencing solar UV composition include geographical location, astronomical factors, pollution levels, climatic conditions such as clouds, and reflective surfaces such as sand, water and snow[18]. In particular, the relative proportion of UVB is remarkably influenced by these factors, such that the UVA/UVB irradiance ratio may vary between 23 and 32 in a typical daylight spectrum[20]. This reflects upon the need for a sensor that can generate a unique response under different combinations of UVR. However, due to the lack of a suitable material that can clearly distinguish between different UVR, the development of spectrally–selective UVR sensors remains challenging.

Most of the current UV sensors rely on the photocurrent generated by an inorganic semiconductor associated with a multicomponent photodetector or spectroradiometer[21–24]. While UVR-sensitive, these expensive equipments require extensive spectral calibration and are not commercially viable for day-to-day consumer-based monitoring. This has seen emerging interest in colorimetric UV sensors that utilize either photochromic organic molecules[25–28], or photocatalyst-mediated dye degradation mechanisms[29,30]. However, in this case, if a low bandgap semiconductor is used, it fails to provide the required spectral selectivity, as above a certain energy threshold, the organic molecules typically convert to a single activated state, irrespective of the type of the UVR[29]. Conversely, the use of a large band gap semiconductor will only allow detection of a specific high energy UVR, but will not respond to other low energy UVRs[30]. A potential option to circumvent these issues is to use band-pass optical filters to separate specific wavelengths of UVR[31]. However, this tends to compromise the sensitivity to an extent that the sensor may not be able to detect the low levels of UV that are practically encountered in day-to-day use. In addition, the associated design complexity along with the use of expensive optical components may make them cost-prohibitive for mass usage.

The current work proposes a new strategy that for the first time, allows spectrally selective colorimetric differentiation of UVA, B, and C by naked eye. We replace broad UVR responsive organic dyes with a multi-redox active photoelectrochromic molecule that generates a unique response to specific UV wavelengths. This offers the required spectral selectivity for efficient UVR monitoring. Using this concept, we are able to develop an invisible ink that can be used to create a low-cost paper-based chromogenic UV sensor by employing simple household items such as filter paper, fountain pen and transparency sheets (Fig. 1). We show that the design flexibility embedded within our technology enables the facile on-demand production of personalized sensors that are suitable for naked-eye UV dosimetry to meet the specific needs of people with different skin phototypes ranging from very fair to dark brown skin (types I–VI) (Supplementary Table 1) [32].

## Results

**Multi-redox photochromic ink for selective UV monitoring**. To assess the ability of multi-redox active photochromic molecules in distinguishing different types of UVRs, we utilized phosphomolybdic acid (PMA) as a representative photoelectrochromic polyoxometalate (POM) of the Keggin family (Fig. 1). Since their discovery in the early 19th century by Berzelius[33], POMs of Keggin anion structure have received widespread attention due to their diverse redox chemistry and rich electromagnetic, catalytic, biological, and photochemical properties[34–45]. A particularly exciting property of POMs is their ability to undergo a multi-step reduction in the presence of an electron ($e^-$) donor and UV light, leading to a family of mixed-valence species[46]. The reduced POMs, commonly known as "heteropoly blues", show a characteristic deep blue color, thereby providing them with photoelectrochromic properties[47,48]. However, to be able to use POMs to spectrally differentiate among different UVRs, a critical step is to select an appropriate $e^-$ donor that is able to reduce PMA molecules to different redox states under different UVRs, particularly in an ambient oxygen-rich environment.

To achieve this, PMA was first exposed to a series of potential organic $e^-$ donors under UVA, B, and C irradiations in an ambient environment. While pristine PMA solution is almost colorless (light yellow) with an absorption maximum at ca. 310 nm, the photoexcited PMA undergoes a metal-to-metal charge transfer in the presence of an $e^-$ donor, and turns into a reduced heteropoly blue with a broad absorption band in the visible region centered at ca. 700 nm (Supplementary Fig. 1)[48]. The highest energy of UVC produces the largest response, followed by UVB and UVA, respectively. Among the various $e^-$ donors, oxalic acid, glycolic acid, and lactic acid (LA) are able to induce differentiable levels of PMA reduction under different UV sources over 30 min (Fig. 2). LA is clearly the most promising candidate both in terms of providing the strongest response (intense blue color suitable for naked-eye detection) as well as high spectral selectivity (differentiation between UVA, B, and C). Inset in Fig. 2 further shows the ability of LA to offer UVR exposure time-dependent reduction of PMA, thereby signifying the potential of PMA–LA system as a colorimetric UV sensor with high spectral selectivity

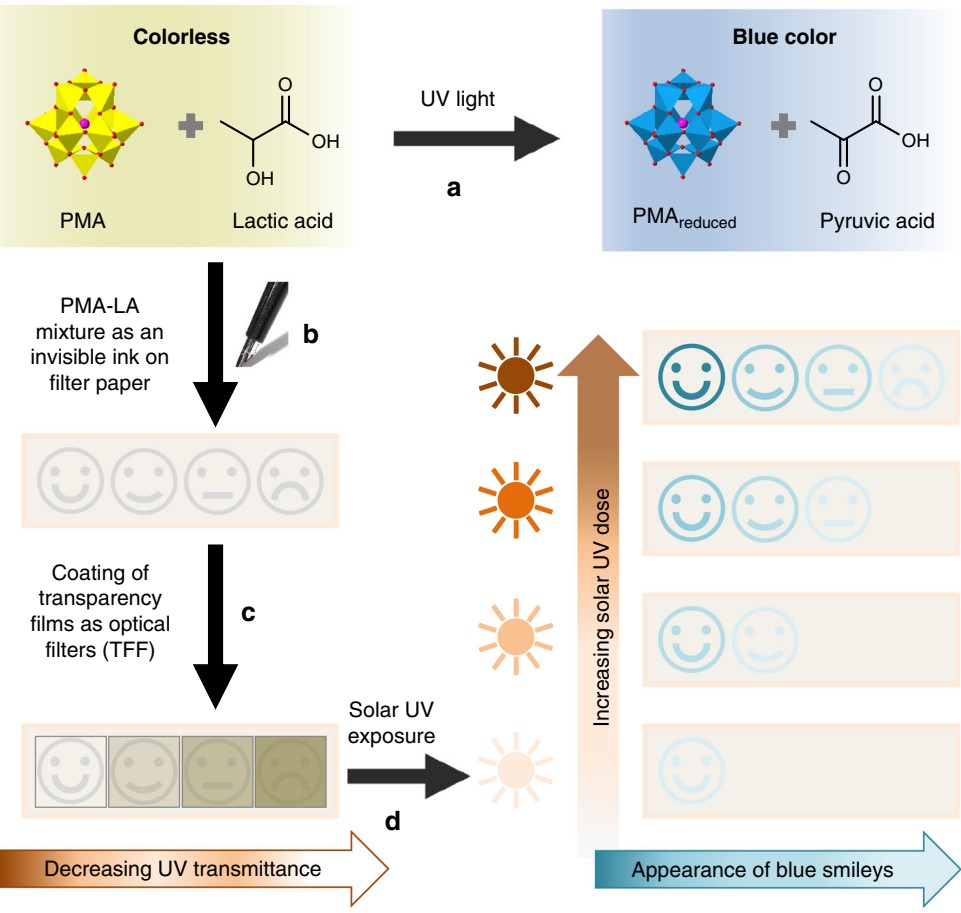

**Fig. 1** Schematic illustration of the colorimetric paper-based UV sensor. **a** The aqueous solution containing photoelectrochromic phosphomolybdic acid (PMA) is reduced by UVR in the presence of e⁻ donor lactic acid (LA) to produce a blue product. **b** The PMA–LA mixture acts as an invisible ink to pen-draw four smileys on a strip of filter paper. **c** These smileys are coated with increasing number of transparency film filters (TFF) that increasingly reduce the UV transmittance on smileys from left to right, thereby allowing the control over color generation time for the individual smiley. **d** After solar UV exposure, blue smileys start to appear on the paper strip sensor from left to right. The number of blue smileys indicates that the user has been exposed to increasingly larger doses of UV, such that the appearance of 1–4 blue smileys represents 25, 50, 75, and 100% of safe exposure thresholds, respectively. The control over the number of TFF over each smiley in the sensor strip allows production of personalized UV sensors specific to different skin phototypes ranging from very fair to dark brown skin, as demonstrated later through Fig. 5

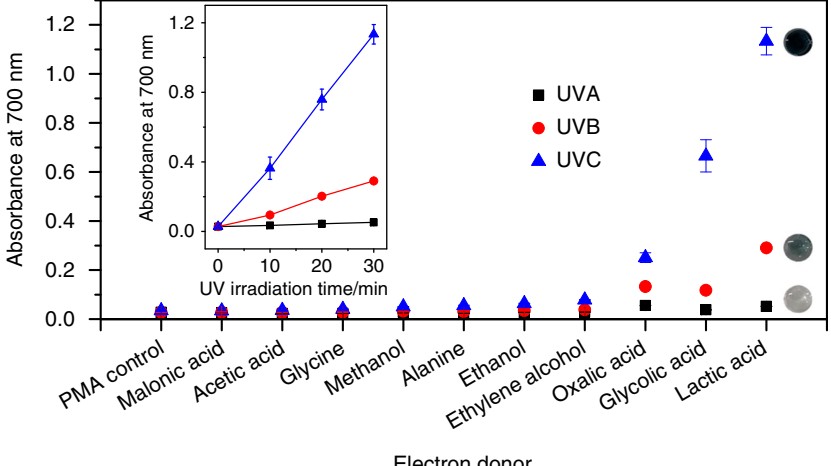

**Fig. 2** Comparison of different e⁻ donors in reducing PMA on UVR excitation. The main panel compares the ability of different e⁻ donors in reducing PMA after 30 min of UVR excitation. The photographs of the colored solutions post-UVR exposure in the presence of lactic acid are shown. The inset shows the UVR exposure time-dependent response of PMA in the presence of lactic acid. The concentrations of PMA and the e⁻ donor correspond to 1 and 10 mM, respectively, and the UVR intensity is consistent as 15 W m⁻² resulting in an effective UV dose of 27,000 J m⁻² over 30 min. Each data point represents an average of the colorimetric response obtained from three independent samples and the associated standard deviation

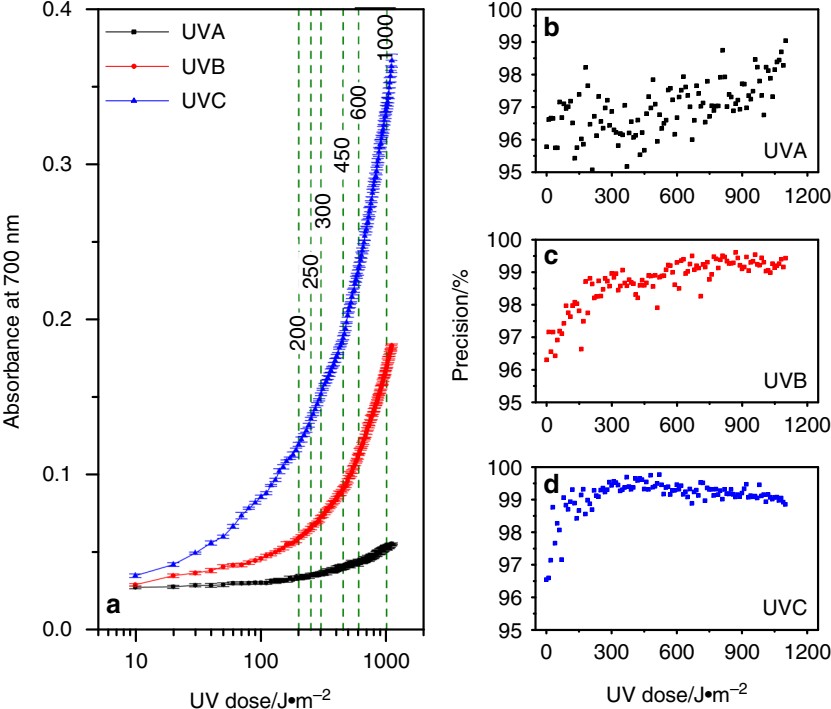

**Fig. 3** Performance of PMA–LA sensor across the UVB MED range for different skin types. **a** UVR dose-dependent colorimetric response of PMA–LA sensor demonstrating the sensor's ability to differentiate UVA, B, and C even at extremely low dosages, as reflected from logarithmic X-axis. Each data point represents an average of the colorimetric response obtained from 12 independent sensors and the associated standard deviation. The highlighted responses correspond to the UVB MED for types I–VI skins (200, 250, 300, 450, 600, and 1000 J m$^{-2}$, respectively). **b–d** The precision of PMA–LA sensors at each of the UV doses at 10 J m$^{-2}$ increments, as calculated from the data presented in **a**

and dose sensitivity. Time-dependent UVR sensor responses in the presence of all ten e$^-$ donors are presented in Supplementary Fig. 2.

Since the intensity of blue color produced on UV exposure is expected to directly depend on PMA concentration, we next compared the sensor performance with increasing PMA concentrations, while also evaluating the role of the relative Molar amounts of the e$^-$ donor LA in achieving the optimum sensor performance (Supplementary Fig. 3). As the PMA concentration is sequentially increased from 1 to 5 mM, the overall colorimetric response continues to increase. However, an interesting trend in the response is observed with respect to the relative Molar quantities of PMA and LA in the sensor. In all cases, irrespective of the UVR source or the PMA concentration, as LA concentration is increased, the absorbance at 700 nm initially rises steeply, which is then followed by a sharp decline. To understand this behavior, we collected the full spectral profiles of the UV-exposed solutions containing a fixed concentration of PMA mixed with increasing concentrations of LA (Supplementary Fig. 4). Spectral analysis reveals that as the amount of the e$^-$ donor with respect to PMA (LA:PMA) increases, the absorption feature at ca. 700 nm due to PMA reduction gradually increases in intensity. However, with a further increase in the e$^-$ donor concentration, an additional absorption peak at ca. 840 nm starts to emerge that continues to dominate along with a concomitant reduction in the absorbance at ca. 700 nm. The observed behavior is attributed to the multi-step photo-electroreduction of Mo(VI) to Mo(V) with an increasing e$^-$ donor concentration that leads to the coexistence of multiple products of distinct optical properties[49]. Specifically, the two reaction products noted at ca. 700 and 840 nm are the outcomes of two successive two-electron reduction steps, each of which allows two of the twelve Mo(VI)

atoms in PMA to be reduced to Mo(V)[50–52]. This highlights the importance of utilizing specific quantities of PMA and LA in developing a colorimetric UVR sensor. Overall, the most intense colorimetric blue response and the highest spectral selectivity toward different UVRs are achieved with 5 mM PMA and 300 mM LA, and as such, this solution was employed as an invisible ink for subsequent studies.

**Ultrasensitive UV sensing for different skin phototypes**. For a UV sensor to be a viable tool to alert users against excessive solar UV exposure, it is critical that the sensor is able to operate in the range of minimal erythemal dose (MED). MED is defined as the threshold dose of UVRs that produces an erythemal reaction, commonly known as sunburn[32,53]. Different skin phototypes have different levels of UV tolerance, and as such, the MED for UVB, which is predominantly responsible for sunburns, varies from 200 J m$^{-2}$ for very fair skin people (type I) to 1000 J m$^{-2}$ for people with dark brown skin (type VI) (Supplementary Table 1)[32]. In contrast, the high tolerance of our skin to UVA (MED in kJ m$^{-2}$) allows safe exposure of over 1000 times higher doses of UVA before causing sunburn. Since the most damaging effects of solar exposure are induced by UVB, we assessed whether the PMA–LA sensor retains its ability to differentiate between UVA, B, and C when exposed to the prescribed UVB MED for different skin types (<200–1000 J m$^{-2}$) (Fig. 3 and Supplementary Fig. 5). The sensor continues to show good spectral selectivity even at low UV dosages, such that the nature of the UVR at as low as 20 J m$^{-2}$ UV dose can be confidently distinguished. The sensor also shows better than 95% precision for all the UV doses tested at 10 J m$^{-2}$ increments. This demonstrates the in-principle feasibility of employing PMA–LA mixture for real-world solar UV monitoring.

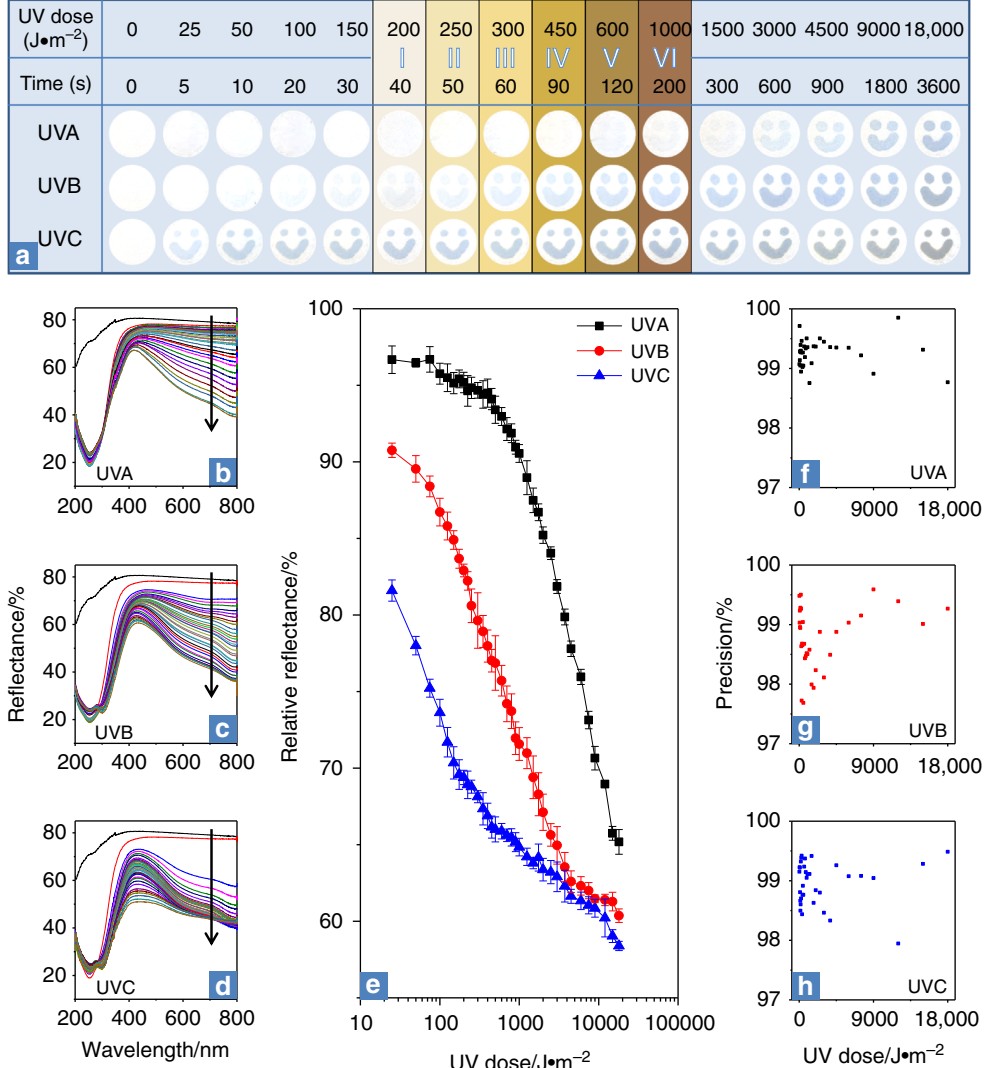

**Fig. 4** The use of PMA–LA invisible ink to fabricate paper-based smiley sensors. **a** Photographs of three paper-based UV sensors with increasing exposure time and the corresponding cumulative effective doses of UVA, B, and C. Only selected time points of photographs are shown for clarity. The sensor response at the UVB MED doses of skin types I–VI is highlighted. The 0 s time point represents invisible smileys drawn on filter papers using PMA–LA ink, followed by drying. **b–d** Reflectance spectra of smileys shown in **a** on exposure to UVA, B, and C, respectively, with increasing UVR doses as indicated by arrows. In comparison to 16 dose points in **a**, reflectance spectra in **b–d** correspond to all the 35 tested doses. **e** UVR dose-dependent colorimetric response of PMA–LA smiley sensors demonstrating the sensor's ability to differentiate UVA, B, and C even at extremely low dosages, as reflected from the logarithmic X-axis. Each data point represents an average of the colorimetric response obtained from four independent sensors and associated standard deviation. **f–h** The precision of PMA–LA smiley sensors at each of the tested UV doses, as calculated from the data presented in **e**

**PMA–LA as invisible ink for naked-eye UV detection on paper.** We next demonstrate the practical applicability of this novel photoelectrochromic PMA–LA system by using it as an invisible ink for designing low-cost, disposable, paper-based UVR smiley sensors (Fig. 1). We used a standard fountain pen to hand-draw smiley faces on three filter paper discs of 15 mm diameter, followed by independently exposing them to UVA, B, and C radiations. The ink is colorless, thus the smileys drawn on the paper are invisible at first (Fig. 4a and Supplementary Fig. 6). Next, as a result of receiving electrons from LA under different wavelengths of UV excitation, PMA in the ink gradually converts into heteropoly blue, resulting in the appearance of blue smileys on the paper whose intensity increases with increasing UV exposure time or total effective dose. It is also clear that the invisible PMA–LA ink retains its ability to differentiate between UVA, B, and C both within and well beyond

the recommended MED of UVB for different skin phototypes. Figure 4b–d compares the corresponding reflectance spectra of these smileys upon exposure to UVA, B, and C, respectively. Comparison of the reflectance spectra of the pristine paper with those after drawing smileys before UV exposure shows strong absorption at ca. 260 nm. This corresponds to the Mo–O charge transfer band in unreduced PMA molecules[48]. Further, as these smiley sensors are exposed to an increasing UV dose, irrespective of the type of UVR, the reflectance in the visible 400–800 nm region concomitantly reduces (Fig. 4e and Supplementary Fig. 7). This reduced reflectance results from the enhanced absorbance of visible light by blue smileys due to the appearance of intervalence charge transfer bands of reduced PMA in the visible region[48]. It is noted that once the paper-based smiley sensors are exposed to a finite dose of UVR, the reflectance and corresponding color of these sensors begin to

saturate. The shortwave UVC saturates the sensor at lower UV doses in comparison to that of UVB and UVA, respectively. This behavior is advantageous considering the high risks associated with UVC in comparison to that of UVB and UVA. Interestingly, while the reflectance profiles of sensors exposed to UVA and B are similar with corresponding blue smileys, the UVC-exposed sensors show a notably different profile with bluish-green smileys. This is attributed to the high energy levels of UVC that can further reduce Mo(VI) atoms in multi-redox PMA to new states with $d$–$d$ band transitions at ca. 500 nm, resulting in different optical properties, as evident from Fig. 4d[48]. These paper-based smiley sensors show outstanding spectral-selectivity with a precision better than 97% for all three types of UVRs across a broad dynamic range of the UV dose. Importantly, the sensors are able to monitor UVA, B, and C with different color changing rates (first-order kinetic rate constants of 0.0012, 0.0136, and 0.0677 s$^{-1}$, respectively, as derived from Fig. 4e), offering UVR spectral selectivity. Relevant to the monitoring of safe UV exposure thresholds on human skin, the smiley paper sensors are able to differentiate between UVA, B, and C when the exposure dose is as low as 25 J m$^{-2}$ or as high as 18,000 J m$^{-2}$. This not only covers the UVB MED range for all skin types, the ability to detect significantly higher UV doses also suggest the potential applicability of these sensors for long-term monitoring of engineered products. As such, the PMA–LA smiley sensors reported here are significantly more sensitive than previously reported paper-based and epidermal UV sensors that are unable to detect UVB MED for some of the skin types through naked-eye dosimetry [29,31].

**Specificity, durability, and stability of PMA–LA UV sensors**. To assess the practical utility of PMA–LA UVR sensors, we studied their robustness through performing a series of specificity, durability and stability studies. As sunlight contains visible and infrared (IR) components in addition to UVR, we first assessed the influence of these wavelengths as well as the heat produced by IR light in potentially causing non-specific sensor response. Under intense visible and IR light irradiation, as well as at an elevated temperature of 80 °C, no change in the reflectance spectra is observed and the smileys remain invisible (Supplementary Fig. 8). This validates the high specificity of PMA–LA sensors toward UVR. Further, these PMA–LA sensors show high durability in the presence of different environmental variants (Supplementary Fig. 9 and Fig. 10). The pre-exposure of these sensors to high intensity visible and infrared radiations, the high ambient temperature of 50 °C and high relative humidity of 90% does not alter their UV dose-dependent colorimetric performance (Supplementary Fig. 9). These sensors are in fact highly robust, and they continue to perform consistently even when they are exposed to a series of humidity/temperature combinations during UV sensing (Supplementary Fig. 10). In addition, the PMA–LA ink employed to prepare smiley sensors remains stable for at least over 8 weeks (Supplementary Fig. 11). In fact, the comparison of all paper-based smiley sensors prepared in this study, including those subjected to rigorous environmental conditions and those prepared using old ink (Fig. 4, and Supplementary Fig. 9-11) showed an outstanding precision of greater than 98% at all the tested doses (inset, Supplementary Fig. 11). In combination, these results clearly demonstrate the high specificity, durability, stability, and reliability of PMA–LA ink and paper-based sensors prepared from these inks, under different storage and in-use conditions.

**Skin type-specific wearable solar UV dosimeters**. As stated earlier, people with different skin colors have different MED thresholds (Supplementary Table 1). Therefore, even though the

sensor presented in Fig. 4 is suitable as an overall UV indicator, it might be challenging to utilize it for obtaining an accurate UV exposure scale (e.g. 25%, 50%, 75%, and 100% exposure thresholds) for people with different skin phototypes. This challenge can be potentially addressed by covering the smiley sensors with optical filters that can alter the UVR dose reaching the smileys, thereby allowing the adjustment of the sensor response time as a function of the UV dose. However, the chosen filter should be cheap enough to facilitate technology adoption by the majority of the population. We discovered that low-cost readily available transparency films can, in fact, act as ideal UV filters for this purpose. Supplementary Fig. 12 shows the ability of different stacks of transparency film filters (TFF) in reducing the light transmittance across different regions of the UV-visible spectrum. Supplementary Fig. 13 verifies that when the smiley UV sensors prepared with different TFF coatings are exposed to simulated sunlight, the colorigenic response is clearly delayed as the number of TFF is increased from 0 to 8. This TFF-induced delayed sensor response is further evident from Supplementary Fig. 14, which shows that the first-order kinetic rate constant of the sensor response of 8 layers TFF-coated sensor (0.0012 s$^{-1}$) is reduced by 4.4 times compared to that from the pristine uncoated smiley (rate constant 0.0053 s$^{-1}$). It is also noted that each of these TFF-coated sensors retains their individual ability to provide UV dosimetry through generating an increasingly higher colorimetric response with the increased exposure time. This suggests that by choosing an appropriate coating of TFF layers, colorimetric UV dosimeters that provide accurate scales for monitoring different levels of solar UV exposure for people with different skin phototypes can be easily customized.

Figure 5 illustrates the sensor design for solar UV dosimetry of people with different skin colors. The sensor comprises smileys drawn on four filter paper discs using a PMA–LA-based invisible ink, and attached to a flexible band. Within each sensor, the individual smileys are covered with different layers of TFF (0–8), such that the number of TFF increases from left to right and on photo-exposure the UV transmittance decreases accordingly. This means that the first smiley will light-up blue with less UV exposure, and increasing larger cumulative UV doses will make other smileys subsequently appear on the paper. Using this concept and through optimization of the number of TFF, we are able to custom-design six personalized sensors that are specific for different skin types ranging from type I (very fair) to type VI (dark brown). In each of these sensors, the first two are happy smileys that show-up at 25% and 50% UVB MED, respectively, when subjected to simulated solar conditions. The third one is a flat smiley representing 75% MED exposure, thereby providing a pre-warning signature of UV exposure threshold. Finally, once the frowning smiley appears, it is a warning that the user has approached the maximum threshold of the safe exposure, and actions must be taken to avoid excessive exposure.

**Discussion**
Our work has shown an innovative concept in spectrally selective dosimetry of different UV irradiations by employing PMA as a multi-redox active photoelectrochromic molecule. This work establishes the unique ability of POMs to spectrally differentiate UVA, B, and C radiations. We first developed an invisible ink by identifying lactic acid as an effective e$^-$ donor that reduces PMA into "heteropoly blues" to varying extents in the presence of different UVRs. This allowed us to validate the proof-of-concept suitability of the PMA–LA ink for spectrally selective detection of UVRs in a solution-based system. This ink was found highly sensitive to allow detection of low levels of UV doses that are typically encountered during a solar exposure event. We further

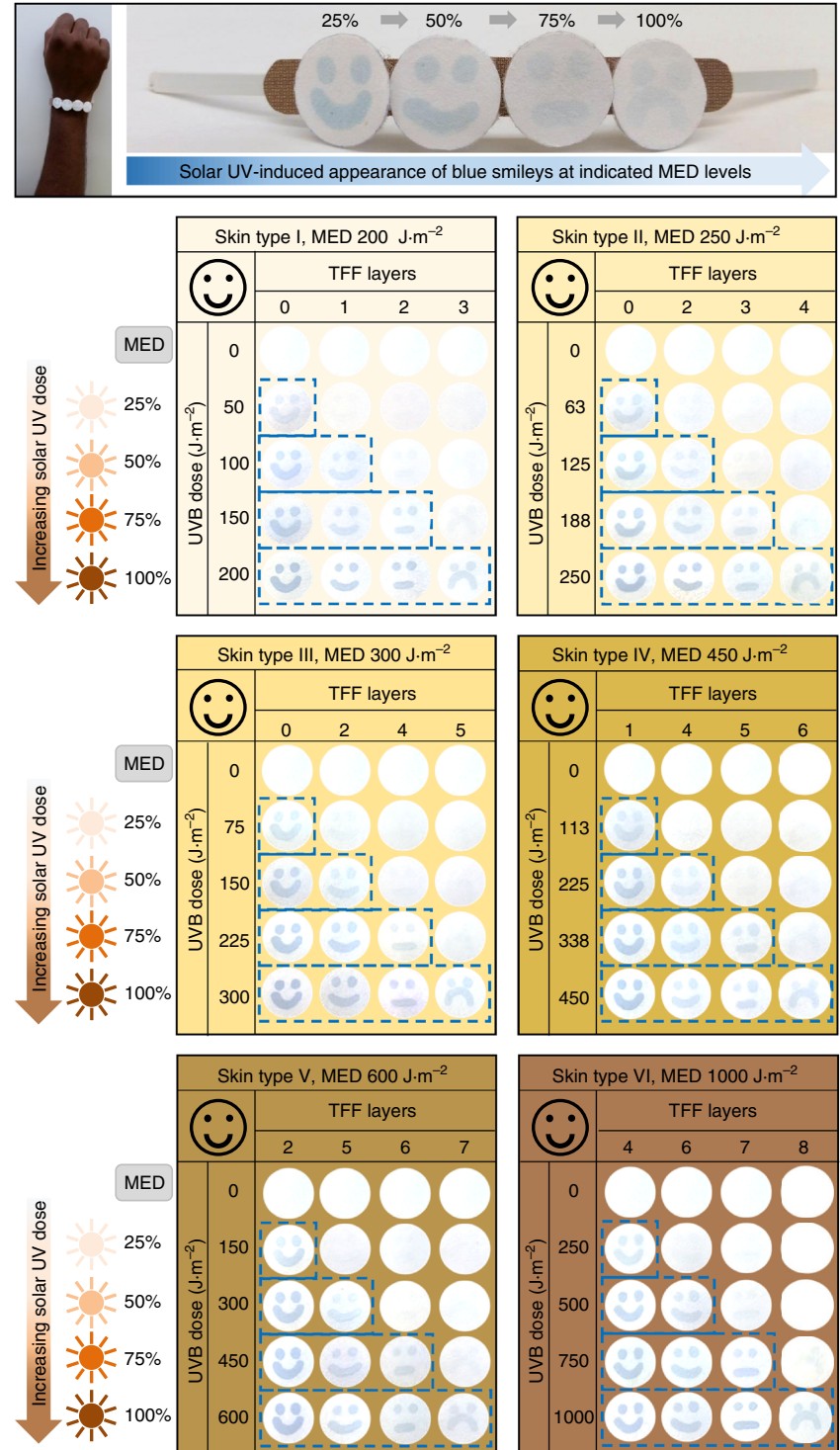

**Fig. 5** Design and performance of skin-phototype-specific personalized UV sensors. The top panel shows a paper-based solar UV sensor prototype for a skin type V person in a wristband format after 100% MED exposure. Sensor design incorporates four paper disc-based smileys, each covered with different layers of TFF, and attached to a flexible wearable band. The smileys are initially invisible, but as the wearer is increasingly exposed to safe solar UVB thresholds of 25%, 50%, 75%, and 100% MED of that particular skin color, the smileys turn blue from left to right, respectively. The colored smileys correspond to total effective solar UV dose. This is demonstrated through assessing the performance of six customized sensors in simulated solar light (bottom six panels), assuring that sensors meet the wide-ranging UVB MED thresholds of people with different skin colors. The skin-phototype specificity is achieved by an appropriate combination of the smiley paper discs coated with different TFF layers (0–8) in a single sensor that allows dose-dependent modulation of sensor response

extended this concept to real-time naked-eye dosimetry of solar UV exposure by designing six personalized paper-based smiley sensors for people with different skin colors. Notably, after a sunburn incident, it may take over 12 h before the signs of the erythemal reaction appear[17]. This makes it extremely challenging for an average person to determine their sun-safe exposure limits. Our sensors generate unambiguous real-time colorimetric warning indications of different UV scales with increasing UVR exposure while providing a reliable sensor response under a variety of environmental conditions. This capability has remarkable potential in empowering users with different skin phototypes to control their maximum sun-safe limits. Importantly, the fabrication of these skin-specific UV dosimeters only requires readily available low-cost components such as filter paper, fountain pen, and transparency sheets. Additionally, the low viscosity and high stability of the invisible ink makes it suitable for various printing options, affording great promise for large-scale fabrication of low-cost paper UV sensors tailored specifically for an individual's need.

## Methods

**Materials and methods.** PMA, LA, glycolic acid, and malonic acid were purchased from Sigma-Aldrich; alanine and glycine were from BDH Chemicals Ltd Poole England; methanol and acetic acid were from VMR International SAS; and ethanol, oxalic acid and ethylene alcohol were from Chem-Supply, Merck Pty Ltd Australia, and May & Baker Ltd Dagenham England, respectively. The filter paper was purchased from Advantec Toyo, and the transparency films (TFF) were obtained from Officeworks Australia. UV-visible absorbance spectra of solutions were recorded in a 96-well plate format using a Perkin Elmer Envision$^{TM}$ 2104 Multi-label Plate Reader. Transmittance and reflectance spectra from the transparencies and filter papers, respectively, were recorded using an Agilent Technologies Cary 7000 UV-Vis-NIR Spectrophotometer equipped with an integrating sphere. The UV light sources comprised two simultaneously used 8-Watt halogen tubes with $\lambda_{max}$ at 365 nm (UVA), 302 nm (UVB) and 254 nm (UVC), respectively. A 300-Watt Ultra-Vitalux Osram lamp was used to simulate the solar spectrum. The working distance between the light sources and the samples was optimized to obtain the required photo-intensity/dose at the sample surface using a Thorlabs PM 200 optical power and energy meter with an S302C sensor. The humidity and temperature measurements were performed using QM7312 Digitech Jumbo Thermometer with Hygrometer & Min Max Memory. Throughout the study, the term "intensity" corresponds to the power of the light source at the surface of the sample with the unit expressed as $W\,m^{-2}$, whereas the term "dose" corresponds to the cumulative dose of a specific radiation on the sample surface over a period of time expressed as $J\,m^{-2}$ $(1\,W\,m^{-2} = 1\,J\,m^{-2}\,s^{-1})$.

**Selection of e⁻ donors for photo-electroreduction of PMA.** A fixed volume of 100 μL of aqueous PMA solution (2 mM) was independently mixed with 100 μL of aqueous solutions (20 mM) of malonic acid, acetic acid, glycine, methanol, alanine, ethanol, ethylene alcohol, oxalic acid, glycolic acid, lactic acid in a 96-well plate. A mixture of 100 μL of PMA solution and 100 μL of $H_2O$ served as PMA control. The plate was then exposed to different UV light sources with a fixed UV intensity of $15\,W\,m^{-2}$ at the sample surface. Time-dependent photochemical reduction of PMA was monitored by measuring the absorbance of the solutions at 700 nm after 10, 20, and 30 min of irradiation. The tests were performed in triplicates, and average absorbance plotted along with standard deviations.

**Optimization of PMA–LA ink for colorimetric sensor.** To optimize the most intense blue response from the PMA–LA mixture in the presence of UVR, concentrations of PMA and LA were varied in a series of experiments. Briefly, 100 μL of PMA solution (2, 5, and 10 mM) was mixed with 100 μL of different concentrations of LA in a 96-well plate to achieve LA:PMA Molar ratios of 0:1, 10:1, 20:1, 30:1, 40:1, 50:1, 60:1, 70:1, 80:1, 90:1 and 100:1, respectively. The plates were then independently exposed to UVA, B, and C with a fixed intensity of $15\,W\,m^{-2}$. The sensor response was obtained by measuring the solution absorbance at 700 nm after 5 min of irradiation, leading to a total UV dose of $4500\,J\,m^{-2}$. The tests were performed in triplicates, and the average response plotted along with the standard deviations (Supplementary Fig. 3). To obtain a mechanistic understanding of the spectral response of different PMA:LA mixtures on UV excitation, the absorption spectra of PMA–LA mixtures containing 1 mM PMA and different LA:PMA Molar ratios were also collected after photoexcitation with UVA for 60 min (Supplementary Fig. 4).

Further, the ability of PMA–LA sensor to produce a colorimetric response in the presence of low doses of UVR that are relevant to the UVB MED for different skin types was evaluated. These experiments involved independently exposing the mixtures containing 5 mM PMA and 300 mM LA (total 200 μL volume) in 96-well

plates, each with 12 replicates, to different UVR with the UV intensity of $5\,W\,m^{-2}$ for different amounts of time leading to different UV doses. This was followed by collecting the time-dependent absorption profile of these samples at 700 nm. The logarithmic data presented in Fig. 3a distinguish early dose points, while the same data is also presented in the linear dose scale in Supplementary Fig. 5. The data represent the mean colorimetric response of 12 independent sensors along with associated standard deviation as error bar at each dose point. The data from these 3960 independent measurements (110 dose points × 3 UVR × 12 replicates) were then used to calculate the sensor precision at each of the tested doses of UVA, B, and C, as presented in Fig. 3b–d.

**Fabrication of paper-based UV smiley sensors.** An aqueous solution containing 5 mM PMA and 300 mM LA was used as an invisible ink for sensor fabrication. To fabricate the paper-based smiley sensors, smileys were hand-drawn on filter paper discs of 15 mm diameter using a standard fountain pen. The smileys were visible as water-marks when the ink was wet, however, the smileys became invisible after drying under ambient conditions while avoiding solar or UV exposure. These dried paper discs were then ready to be used as paper-based UV sensors shown in Fig. 4.

**Photo-exposure experiments on paper-based smiley sensors.** The hand-drawn paper-based smiley sensors were independently exposed to UVA, UVB, and UVC with a fixed intensity of $5\,W\,m^{-2}$. Photographs of the paper-based sensors were taken using a mobile camera before and after UV irradiation for different time points (0, 5, 10, 15, 20, 25, 30, 35, 40, 45, 50, 60, 70, 80, 90, 100, 120, 140, 160, 180, 200, 250, 300, 350, 400, 500, 600, 750, 900, 1200, 1500, 1800, 2400, 3000 and 3600 s), resulting in the total effective UV dose of $25–18,000\,J\,m^{-2}$. Supplementary Fig. 6 shows corresponding photographs as-obtained from the mobile camera. Since, after photo acquisition, the acquired color contrast was found to be different than the original white background of the paper-based sensor perceived by the naked-eye, the background of the photos was changed to white using the brightness and contrast correction tool of Microsoft PowerPoint 2010. Same setting parameters were applied to all the images to retain consistency. These edited photos that appeared closest to those visually perceived by the naked-eye are presented in Fig. 4a.

The color intensity of the blue smileys appearing on the paper with increasing UVR exposure was quantified by recording their absolute reflectance. The reflectance of the blank filter paper without drawing served as control (Fig. 4b–d). The relative reflectance of the smiley sensors under different photoexcitation conditions was calculated by integrating the area under each of the reflectance curves in the 400–800 nm range while considering the reflectance from the blank untreated paper as 100%. The logarithmic data presented in Fig. 4e distinguish early dose points, while the corresponding data are also presented in a linear dose scale in Supplementary Fig. 7. The data correspond to the mean colorimetric response from four independent sensors along with associated standard deviation as error bars at each of the dose points. The data from these 420 independent measurements (35 dose points × 3 UVR × 4 replicates) were then used to calculate the sensor precision at each of the tested doses of UVA, B, and C (Fig. 4f–h).

**Specificity, durability, and stability of PMA–LA UV sensors.** The specificity of paper-based smiley sensors was assessed by exposing the sensors independently to high intensity visible light ($130\,W\,m^{-2}$ at the sensor surface), infrared light ($700\,W\,m^{-2}$ at the sensor surface), and 80 °C oven heat, followed by collecting their reflectance spectra (Supplementary Fig. 8).

To validate the durability of paper-based smiley sensors, they were pre-exposed for 1 h to a variety of conditions mimicking ambient environment, followed by exposure to different doses of UVB, and collecting their reflectance profile (Supplementary Fig. 9). These pre-exposure conditions included high intensity visible light ($130\,W\,m^{-2}$), high intensity IR light ($700\,W\,m^{-2}$), the high ambient temperature of 50 °C, and high relative humidity of 90%. Additional durability tests mimicking different climatic conditions involved assessing the UVB dose-dependent sensing performance of paper-based smiley sensors from reflectance studies, while keeping them continuously exposed to varying relative humidity/temperatures during sensing (Supplementary Fig. 10).

To establish the stability of the UV sensing platform, we tested the stability of the PMA–LA ink over 8 weeks (Supplementary Fig. 11). This was done by preparing the PMA–LA ink at the 8th, 6th, 4th, 2nd, and 0th week, followed by fabricating paper-based smiley sensors using these inks at the 0th day, and comparing their ability to detect different doses of UVB through reflectance studies (Supplementary Fig. 11).

**Controlling UV dosimetry using transparency film filters (TFF).** To modulate the amount of solar irradiation reaching the surface of paper-based smiley sensors, standard transparency sheets were employed as low-cost optical filters (TFF). To validate the utility of these transparency films as optical filters, the initial experiments involved collecting the transmittance profile of different stacks of TFF (1–8 layers) (Supplementary Fig. 12). This was followed by fabrication of paper-based smiley sensors, and covering them with different layers of TFF (0–8 layers) before exposing to a solar simulator with a consistent UVB intensity of $1\,W\,m^{-2}$. The blue color appearing on the paper sensors was quantified by monitoring their

reflectance profile after the cumulative exposure time of 30, 60, 120, 180, 300, 600, and 900 s, corresponding to the respective effective UVB dosage of 30–900 J m$^{-2}$ (Supplementary Fig. 13). The relative reflectance of sensors under different TFF coverage is presented in Supplementary Fig. 14.

**Fabrication of skin-specific personalized UV dosimeters**. To fabricate the paper-based solar UV dosimeter, four filter paper discs of 15 mm diameter were taken and different faces (two with the smiling faces, one flat face and one frowning face) were drawn on discs using a fountain pen containing the PMA–LA invisible ink. This was followed by covering each of the smiley-containing discs with a different number of TFF, ranging from 0 to 8. The number of TFF increased from smiling face toward the frowning face. These four smiley discs were attached to a flexible band in a manner such that the disc with the lowest number of TFF is placed on the left-most edge (smiling face) while the disc with the highest number of TFF is placed on the right-most edge (frowning face). The final solar UV dosimeter was in the form of a strip sensor consisted of four invisible smileys coated with different TFF layers, such that on solar exposure, the smileys from left to right will begin to appear blue with the increasing cumulative UV exposure time/dose.

To custom-design six different UV dosimeters specific to different skin types (Supplementary Table 1), the approach described above was repeated; however, in this case, the TFF coating-dependent sensor reflectance data obtained from Supplementary Fig. 13 and Fig. 14 were used as the reference standard to decide the number of TFF layers coatings on each smiley. The number of TFF in each of these UV dosimeters was chosen such that the sensor provides a naked-eye colorimetric dosimetry of 25%, 50%, 75%, and 100% equivalent of UVB MED for each of the skin types. This sensor design allowed appearance of the first smiley at 25% MED, the additional second smiley at 50% MED, and the subsequent appearance of the third and the fourth smileys at 75% and 100% MED, respectively. As such, the layers of TFF coated on four smileys for fabricating personalized dosimeters for each skin types included: 0, 1, 2, and 3 layers for skin type I (Very fair skin: Northern European and British descent); 0, 2, 3, and 4 layers for skin type II (Fair skin: European and Scandinavian descent); 0, 2, 4, and 5 layers for skin type III (Medium skin: Southern European and Central European descent); 1, 4, 5, and 6 layers for skin type IV (Olive skin: Mediterranean, Asian and Latino descent); 2, 5, 6, and 7 layers for skin type V (Brown skin: East Indian, African and Native American descent); 4, 6, 7, and 8 layers for skin type VI (Dark brown skin: African Aboriginal descent).

To perform UV dosimetry experiments under simulated sunlight conditions, the prepared sensor strips for each skin type were exposed to a solar simulator with a consistent UVB intensity of 1 W m$^{-2}$ at the sensor surface for different exposure time. The appearance of blue smileys on photo-exposure allowed naked-eye dosimetry of UV irradiations, as indicated through corresponding photographs recorded using a mobile camera (Fig. 5).

## Data availability

All data are available in the main text or through online supplementary information. Corresponding unprocessed data are available from the corresponding authors.

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

## Acknowledgements

Authors acknowledge the Australian Research Council (ARC) for supporting this work through Future Fellowship (FT140101285—V.B.) and Discovery (DP170103477—V.B.) and R.R.) grants. V.B. acknowledges the generous support of the Ian Potter Foundation in establishing Sir Ian Potter NanoBioSensing Facility at RMIT University. W.Z. acknowledges RMIT University for the Vice Chancellor's postgraduate scholarship; R. R. and S.W. acknowledge RMIT University for Vice Chancellor's Postdoctoral Fellowships; M.B. thanks ARC for a DECRA Fellowship. J.M.D-V. acknowledges the Commonwealth of Australia for an Endeavour Executive Award towards a sabbatical at RMIT University.

## Author contributions

V.B. and J.M.D-V. designed the study. W.Z. performed most the experiments with the help of other co-authors (A.G. participated in screening electron donors; D.J. and R.R. participated in reflectance measurements; M.T., S.W., S.S. and M.B. participated in UV dosimetry). W.Z. processed data. W.Z. and V.B. critically analyzed the data and wrote the manuscript.

## Additional information

**Competing interests:** The work presented in this study has been patented (US 62/684,045). The authors have interests in developing a commercial solar monitoring sensor from the results presented in this study.

