## [Peer Review File · Nature Communications]

Reviewers' comments:

Reviewer #1 (Remarks to the Author):

The authors present the development of UV sensitive dyes and their application in personal UV exposure monitoring. The authors use PMA based chemistry to facilitate spectrally selective and selective UV sensitive compounds. Color response can be tailored by attenuating dose with filters to match physiologically relevant doses for individual skin types.

The manuscript is written well, and the sensors are characterized extensively towards their sensitivity. The also the selectivity of the sensors appears suitable for the application.

However, the authors do not compare their work against existing literature that is very similar and covers more practical use cases and integrates multifunctionality.

H. Araki et al (10.1002/adfm.201604465) published in 2016, for example show a device that not only epidermal embodiment that allows the test subject to wear the device in day by day activities but also provides information on temperature and is engineered for a smart phone readout to extract information on UV A and B dose as well as temperature which are vital parameters to detect skin health.

Similar work has also been commercialized by L'Oreal, (myUV patch) which has the same features presented in this work.

Given this benchmark the authors have to demonstrate significant advance over this technology to justify publication in Nature Communication, which is not the case.

Specific comments and questions to the authors include:

- UVC is largely blocked by the ozone layer – and this seems to be the unique sensing aspect that differentiates this work from prior published papers. The authors should justify, more clearly, how many individuals or what percentage of countries would need a UVC sensor.
- The devices are paper based, this means that sweat and other body fluids will permeate the device. This is especially relevant during exercise in sunny environments. How will such fluids influence device functions? What about ambient humidity?
- Phosphomolybdic acid also has adverse health effects, how are the authors mitigating the risk of leaching, especially considering the paper-based platform.
- Authors demonstrate that PMA-LA ink is stable under visible and IR irradiation, as well as at 80°C. What is the projected lifetime of the UVR sensitive ink proposed to be contained in a fountain pen to create disposable colorimetric UV sensors?
- Authors claim spectral selectivity of the PMA-LA sensors by evidencing different color changing rate under independent UVA, UVB and UVC irradiation at 15 W/m². However, in the case of solar irradiation outdoors, composition of UVA is nearly 99% compared to 1% UVB and almost negligible UVC. It is of interest to understand how spectral selectivity of the PMA-LA sensors may be applied in day-to-day UVR monitoring.
- The other question is the actual measurement precision of these sensors: the author's present J/m² in 50 to 150 J/m² increments. Compared to previously published wearables and existing gold standard

radiometers, this would represent a significant limitation. I suggest the author's provide some sense of the measurement precision in J/m² for UVC, UVB and UVA.

Reviewer #2 (Remarks to the Author):

The authors provide a highly original and novel approach for designing low-cost, high selectivity UV A, B, C sensors based on an innovative ink containing redox-active POMs and an electron donor. The study could significantly develop the field further and is therefore suitable for publication in Nature Commun, as it highlights how readily available materials can be fabricated into a synergistic composite for UV detection.

The manuscript is well prepared, there are only few minor changes which I would suggest before publication is possible.

1. in all UV-Vis spectra: change the unit of absorbance from [a.u.] to [-] as Absorbance is dimensionless, but the numbers are not arbitrary (see Beer-Lambert-law!)
2. Scheme 1: Please arrange the lactic acid molecule so that the OH and carbonyl O point on the same direction, so it is clear what is being oxidized on first sight.

Reviewer #3 (Remarks to the Author):

The designed real-time solar UV dosimeters consisting of a multi-redox polyoxometalate and electron donor was fabricated, and their UV responsibilities were investigated in this manuscript. Overall, the experimental result of this manuscript is well prepared to support the author's claim. Some questions and concerns are raised to improve the quality of the manuscript as follows.

1. The authors claimed that "the current work proposes a new strategy that for the first time, allows spectrally-selective colorimetric differentiation of UVA, B, and C by naked-eye" However, it seems that the recent work "A Zero-Power, Low-Cost Ultraviolet-C Colorimetric Sensor Using a Gallium Oxide and Reduced Graphene Oxide Hybrid via Photoelectrochemical Reactions, Catalysts 7, 248 (2017)" also demonstrated a similar colorimetric differentiation of UVA, B, and C by naked-eye. Thus, it is recommended to refer the relevant work and compared it with the submitted manuscript.
2. The authors presented the results using either reflectance or absorbance in this submitted manuscript. Is there any reason to present the results in that way?
3. There are some typos in the Fig. 3. (a), (b), (c) and (d) were not defined in Fig. 3. Check the caption on the Fig. 3.
4. Though the authors claimed that "this is the first time that POMs, which were in fact discovered almost 200 years ago, have used to spectrally differentiate UVA, B, and C radiations", the differentiation between UVA and UVB seems to be subjective by the naked-eye. Thus, it is recommended to tone down the claim and revise the manuscript accordingly.

We thank the reviewers for their time and critical assessment of our work, which has assisted in further improving the quality of this manuscript. Detailed response to the reviewers' comments is provided below:

Response to Reviewer #1

Overall comment: The authors present the development of UV sensitive dyes and their application in personal UV exposure monitoring. The authors use PMA based chemistry to facilitate spectrally selective and selective UV sensitive compounds. Color response can be tailored by attenuating dose with filters to match physiologically relevant doses for individual skin types.

The manuscript is written well, and the sensors are characterized extensively towards their sensitivity. The also the selectivity of the sensors appears suitable for the application.

However, the authors do not compare their work against existing literature that is very similar and covers more practical use cases and integrates multifunctionality.

H. Araki et al (10.1002/adfm.201604465) published in 2016, for example show a device that not only epidermal embodiment that allows the test subject to wear the device in day by day activities but also provides information on temperature and is engineered for a smart phone readout to extract information on UV A and B dose as well as temperature which are vital parameters to detect skin health.

Similar work has also been commercialized by L'Oreal, (myUV patch) which has the same features presented in this work.

Given this benchmark the authors have to demonstrate significant advance over this technology to justify publication in Nature Communication, which is not the case.

Overall response: Thank you for finding the manuscript well-written supported by extensive characterisation along with appropriate selectivity for the demonstrated application. The reviewer's concern is that we didn't compare our work with existing similar literature, particularly that by H. Araki et al (10.1002/adfm.201604465), that to our understanding might have contributed to development of L'Oreal's myUV patch.

In response, in our original Cover Letter and justification to *Nature Communication* Editors, we had submitted a detailed comparison of our work with the above publication and two additional publications from John Rogers' group at the University of Illinois at Urbana-Champaign. We understand that the reviewer might not have had access to our justification provided to the editorial office. We have now provided the excerpts from our original justification as below:

*"To the best of our knowledge, the closest work that to some extent allows differentiation between UVA and B is by John Rogers Group at University of Illinois at Urbana-Champaign (**Adv. Funct. Mater.**, 2016, DOI: 10.1002/adfm.201604465). Notably, even this work does not offer differentiation of UVC from the above two UV radiations. This study involved use of bandpass filters to achieve selective blocking of UVA vs. UVB during dye activation, and then combining this system with a complex NFC-integrated chip for mobile phone-based monitoring. However, the relatively low sensitivity of this sensor makes it unsuitable for naked-eye solar UV exposure monitoring. For instance, Figures 2 and S3 in this publication show nearly identical visual sensor response under 0 or 500 J/m² UVB exposure. Considering the low solar UVB exposure thresholds for different skin types, the outcomes of this study are unlikely to offer practical significance. To circumvent this issue, the authors have now shifted focus on total solar UV detection instead of spectrally-selective UVA and B differentiation (**Sci. Adv.**, 2016, DOI: 10.1126/sciadv.1600418; **PLoS One**, 2018, DOI: 10.1371/journal.pone.0190233). These NFC-integrated complex optoelectronic devices remain expensive."*

In addition to our above original justification, it is noted that one of these publications (**PLoS One**, 2018, DOI: 10.1371/journal.pone.0190233) details the design and performance of L'Oreal's myUV patch. As reviewer rightly points out, the major emphasis of above reports is to develop NFC-integrated multifunctional optoelectronic epidermal devices.

In contrast, the focus of our current study is on developing the novel chemistry of a spectrally-selective UV ink. This is a significant advance over prior art, as previous reports on UV detection (including the above publications) rely on a single step dye activation/inactivation process. Our

current report on the ability of the multi-redox active polyoxometalates to responding differentially against UVA, B and C radiations will provide a new generic candidate to be integrated with different forms of UV sensors.

Further, the major advantages of the PMA-based paper sensors proposed in our study over those indicated above are:

- (i) Ease in fabrication due to a simple printable ink
- (ii) High sensitivity allowing naked-eye dosimetry without the requirement of a technological interface such as NFC or mobile camera or an external energy source
- (iii) High spectral selectivity
- (iv) Customisable sensors for people with different skin types
- (v) Robust sensors with high specificity, durability and stability across different environmental conditions

It is further noted that we had cited the above relevant reports in the original manuscript and justified the value proposition of our current approach in context of these prior art (last paragraph of Introduction). These reports either show compromised sensitivity or selectivity or both. In particular, the data presented in these publications suggest that these sensors will not be able to detect the levels of UV that are practically encountered in day to day use, especially for skin types I to III. We have now edited the original text to more clearly outline the differences between previous studies and our current work (Lines 60-74).

Comment 1: UVC is largely blocked by the ozone layer – and this seems to be the unique sensing aspect that differentiates this work from prior published papers. The authors should justify, more clearly, how many individuals or what percentage of countries would need a UVC sensor.

Response 1: We agree that UVC is almost completely blocked by the ozone layer, and it is unlikely that day-to-day solar monitoring will require a UVC sensor. We originally mentioned the need for a UVC sensor in Introduction. We have now further elaborated on the importance of UVC monitoring in the revised text in lines 44-49.

As such, it is difficult to estimate how many individuals would need a UVC sensor, but the increasing consumer perception to keep their environment clean, associated with current open and low-cost access to UVC lamps means that more and more users are likely to be exposed to UVC radiations. This reflects upon the importance of the proposed spectrally-selective UV sensor ink.

Comments 2 and 3: The devices are paper based, this means that sweat and other body fluids will permeate the device. This is especially relevant during exercise in sunny environments. How will such fluids influence device functions? What about ambient humidity?

Phosphomolybdic acid also has adverse health effects, how are the authors mitigating the risk of leaching, especially considering the paper-based platform.

Responses 2 and 3: In response, the concept demonstrated in this manuscript is aimed towards a scientific advance that shows the proof-of-the-concept ability to develop a UV-active ink to find applications in paper-based sensors.

For commercial product development, the paper-based UV sensor can be easily housed in a water-proof laminated assembly to avoid its exposure to sweat and other body fluids. This will make it unlikely that the external fluids will influence device functions. Similarly, the potential acidity of phosphomolybdic acid will not lead to adverse health effects, as it will be sealed inside the water-proof assembly.

Notably, since the paper-based UV sensor shown in our current work is not an epidermal sensor, it offers diverse wearable formats, such as wearing on the wrist in the form of a wristband, wearing on the hair as a headband, or potentially wearing on top of the clothes or hats. Considering that a number of people have highly photosensitive skin (particularly those suffering from Lupus and

others on certain medications), in such specific cases, epidermal sensors that stick on the skin may not be as convenient as multi-format wearable sensors.

For the reasons outlined above, the ambient humidity is also unlikely to influence the sensor performance. However, considering that geographical locations and seasons differ in ambient humidity and temperature, we have now performed additional experiments to investigate the influence of a range of ambient relative humidity (RH)/ temperature combinations on the UV sensor performance (Fig. S10, discussion in S-VIII in supporting information, and lines 221-241 in the main manuscript). In these experiments, the paper-based sensors were not housed in a water-proof jacket and their response to different doses of UVB in different RH/ temperature environments was studied. The data shows that the changes in RH/ temperature do not influence the performance of paper-based UV sensors.

Comment 4: Authors demonstrate that PMA-LA ink is stable under visible and IR irradiation, as well as at 80°C. What is the projected lifetime of the UVR sensitive ink proposed to be contained in a fountain pen to create disposable colorimetric UV sensors?

Response 4: Thank you for this important question that is critical for the commercial success of the proposed UV sensor technology. Over the past several weeks, we have performed a series of new experiments to establish the robustness (specificity, durability and stability) of the UV-active ink as well as the paper-based UV sensors.

Briefly, in our original manuscript, we showed the specificity of the sensors through demonstrating that the sensors did not respond to visible and IR radiation and to heat, but responded only to the UVR. However, we didn't show that the UV-active ink and the paper-based UV sensors will continue to perform after their prior exposure to different environmental conditions.

Our new experiments to investigate the robustness of these sensors involved studying:

- (i) the durability of paper-based sensors towards UV sensing after their prior exposure to potential environmental variants including (Fig. S9):
 - a. high-intensity visible light
 - b. high-intensity IR light
 - c. ambient heat of 50 °C
 - d. relative humidity of 90%
- (ii) the durability of the paper-based sensors through assessing their UV sensing performance while exposing them in-parallel to different climatic conditions with a series of humidity/ temperature combinations, as discussed in previous response #2 and #3 (Fig. S10), and
- (iii) the stability of the UV-active ink over 8 weeks through preparing the PMA-LA ink at the 8th, 6th, 4th, 2nd and 0th week; and creating paper-based sensors using these inks at the 0th day, followed by comparing their ability to detect different doses of UVB (Fig. S11).

These studies clearly demonstrate the high specificity, durability and stability of PMA-LA ink and paper-based sensors prepared from these inks, under different storage and in-use conditions (Figures S9-S11, Section S-VIII in supporting information, and lines 221-241 in the main manuscript).

Comment 5: Authors claim spectral selectivity of the PMA-LA sensors by evidencing different color changing rate under independent UVA, UVB and UVC irradiation at 15 W/m². However, in the case of solar irradiation outdoors, composition of UVA is nearly 99% compared to 1% UVB and almost negligible UVC. It is of interest to understand how spectral selectivity of the PMA-LA sensors may be applied in day-to-day UVR monitoring.

Response 5: The composition of relative intensities of UVA and UVB in the solar irradiation falling on the surface of the earth is highly variable. As such, while the intensity of UVA irradiation at a particular geographical location is nearly consistent during the day, the UVA/UVB irradiance ratio may vary between 23 to 32 in a typical daylight spectrum (i.e. ~40% variability) in the case of outdoor solar irradiation. This high variability in the UV irradiation is because UVB irradiation can

be influenced by several factors, including the geographical location, astronomical factors (sun elevation), levels of aerosols and pollution in the environment, reflective surfaces, as well as climate conditions. For instance, clouds may selectively block UVB without influencing UVA. Similarly, under indoor conditions (e.g. houses and vehicles), most of the UVB is blocked and UVA is accumulated. This reflects upon the need for a sensor that can generate a spectrally-selective response for different combinations of UVA and UVB.

Further, not only the composition of UVR is highly variable, the biological effects induced by UVA vs UVB also differ remarkably from one condition to the other. For instance, UVB is more damaging to humans, leading to 100% contribution towards photo-conjunctivitis and 77% contribution towards erythema; whereas UVA is more damaging to phytoplankton leading to UVB contributing ~21% to plant damage.

We already discussed the importance of UVC monitoring in response to the above Comment #1. As such, a spectrally-selective UV sensor that provides quantifiable response under different UV irradiations not only offers opportunity to monitor different health conditions, but also plant health and other industrial processes.

Please refer to our revised text in lines 50-59 that captures the above aspects.

Comment 6: The other question is the actual measurement precision of these sensors: the author's present J/m² in 50 to 150 J/m² increments. Compared to previously published wearables and existing gold standard radiometers, this would represent a significant limitation. I suggest the author's provide some sense of the measurement precision in J/m² for UVC, UVB and UVA.

Response 6: Thank you for the comment. Firstly, we will like to mention that in our original submission, we showed the sensitivity resolution of 15 J/m² for the UV-active ink in a solution format, and 50 J/m² for paper-based sensors (original Fig. 2 for solution-based sensor, and Fig. 4 for Type I skin paper-based sensor). The resolution of solution-based sensor might have got misinterpreted based on the originally highlighted doses in Fig. 2 that were meant to indicate the doses corresponding to Skin types I-VI and indeed showed a difference of 50 J/m². Similarly, for paper-based sensors, the reviewer might have considered the resolution shown in original Fig. 3 which reflected 150 J/m² intervals; whereas sensor for Type I skin in original Fig. 4 reflected 50 J/m² interval resolution.

Further, we have performed extensive literature survey on previously published wearable and similar sensors. However, we could not find any publication that matches the resolution shown in our original submission. For instance, please refer to our response to your overall first comment. We do agree that gold standard radiometers offer much better resolution. However, such radiometers are not suitable for point-of-care monitoring of UV exposures. For solar UV monitoring, the UVB MED exposure threshold for different skin types range from 200 to 1000 J/m². Therefore, the originally shown dose sensitivity was suitable for this particular application.

We, however, highly appreciate your suggestion, as we did not originally calculate the precision of our sensors. This is a critical sensor parameter that must be determined to reflect the robustness of any sensor. Therefore, we have now performed a series of extensive experiments with enhanced number of UV dose points and several replicates, both for solution-based and paper-based UV detection systems. The original Fig. 2 and 3 have been replaced with the new data that shows sensitivity resolution of 10 J/m² for solution-based sensors (lines 167-170), and 25 J/m² for paper-based sensors (lines 209-220).

As such, for reliable calculation of precision, the number of replicates is extremely important. Since solution-based UV monitoring allows multiplexing, the new data shown in Fig. 2 and S5 is based on 3,960 independent measurements comprised of 110 dose points, each with 12 replicates, and each for UVA, B and C. However, the quantification of paper-based UV monitoring requires time-intensive reflectance measurements. Therefore, the new data in Fig. 3, S6 and S7 for paper-based sensors is from 420 independent measurements, comprised of 35 dose points, each with 4 replicates, and each for UVA, B and C monitoring.

This has been an extremely fruitful undertaking, based on which, we have now been able to calculate the precision for both solution- and paper-based detection systems. The precision of the proposed UV sensors at each of the tested doses for UVA, B and C are exhibited in the new Fig. 2 and 3 for solution-based and paper-based sensors, respectively. As such, both solution- and paper-based sensor show better than 95% precision at all the tested UV doses.

Further, based on the sensor response obtained from the robustness study of PMA-LA ink over several weeks, and that of paper-based sensors across a diverse range of environmental conditions (Fig. S9 to S11), we have been able to calculate the overall precision of the paper-based smiley sensors. These sensors show a precision better than 98% across all the tested doses, which establishes confidence in their performance.

Response to Reviewer #2

Overall comment: The authors provide a highly original and novel approach for designing low-cost, high selectivity UV A, B, C sensors based on an innovative ink containing redox-active POMs and an electron donor. The study could significantly develop the field further and is therefore suitable for publication in Nature Commun, as it highlights how readily available materials can be fabricated into a synergistic composite for UV detection.

The manuscript is well prepared, there are only few minor changes which I would suggest before publication is possible.

Overall response: Thank you for identifying the novelty and innovation in this work. The response to the minor comments is provided below.

Comment 1: In all UV-Vis spectra: change the unit of absorbance from [a.u.] to [-] as Absorbance is dimensionless, but the numbers are not arbitrary (see Beer-Lambert-law!)

Response 1: Thank you for raising a valid point. We agree and have therefore removed “a.u.” from all the relevant graphs in our revised manuscript.

Comment 2: Scheme 1: Please arrange the lactic acid molecule so that the OH and carbonyl O point on the same direction, so it is clear what is being oxidized on first sight.

Response 2: We have changed Scheme 1, as suggested.

Response to Reviewer #3

Overall comment: The designed real-time solar UV dosimeters consisting of a multi-redox polyoxometalate and electron donor was fabricated, and their UV responsibilities were investigated in this manuscript. Overall, the experimental result of this manuscript is well prepared to support the author’s claim. Some questions and concerns are raised to improve the quality of the manuscript as follows.

Overall response: Thank you for providing suggestions to further improve the quality of this work. The response to specific comments is provided below.

Comment 1: The authors claimed that “the current work proposes a new strategy that for the first time, allows spectrally-selective colorimetric differentiation of UVA, B, and C by naked-eye” However, it seems that the recent work "A Zero-Power, Low-Cost Ultraviolet-C Colorimetric Sensor Using a Gallium Oxide and Reduced Graphene Oxide Hybrid via Photoelectrochemical Reactions, Catalysts 7, 248 (2017)” also demonstrated a similar colorimetric differentiation of UVA, B, and C

by naked-eye. Thus, it is recommended to refer the relevant work and compared it with the submitted manuscript.

Response 1: Thank you for bringing this paper to our attention. We acknowledge that this paper shows the ability of gallium oxide-rGO hybrid to selectively detect UVC without being responsive to UVA. However, we also note that this study did not investigate the influence of UVB. As such, this work exploits a well-established photocatalyst-mediated dye degradation mechanism, a concept that is well-known and was discussed along with citation of relevant papers in our Introduction. While we cited other papers based on this mechanism, we have now included this additional reference, and discussed it in the context of our current work (Ref 30, lines 60-74).

Comment 2: The authors presented the results using either reflectance or absorbance in this submitted manuscript. Is there any reason to present the results in that way?

Response 2: The data on solution-based UV sensors were presented as absorbance, whereas the data on paper-based sensors were presented as reflectance. This presentation is based on the common practice in the field that while reporting optical properties of opaque (optically non-transparent) substrates, Absolute Reflectance (%) is typically preferred.

Comment 3: There are some typos in the Fig. 3. (a), (b), (c) and (d) were not defined in Fig. 3. Check the caption on the Fig. 3.

Response 3: We apologise for the oversight. The Figure 3 has been replaced with new data, and figure caption has been appropriately modified.

Comment 4: Though the authors claimed that “this is the first time that POMs, which were in fact discovered almost 200 years ago, have used to spectrally differentiate UVA, B, and C radiations”, the differentiation between UVA and UVB seems to be subjective by the naked-eye. Thus, it is recommended to tone down the claim and revise the manuscript accordingly.

Response 4: As suggested, we have now replaced the above statement with “This work establishes the unique ability of POMs to spectrally differentiate UVA, B and C radiations.” (lines 310-311).

REVIEWERS' COMMENTS:

Reviewer #1 (Remarks to the Author):

The authors provide some useful responses to reviewer comments, but my original judgement that this manuscript does not rise to a level required for a publication in Nature COmmunication remains. The basic concept of using color change chemistries to monitor UV exposure in a thin flexible device are already in the literature (for UVA and for UVA and UVB separately), and they form the basis for several commercially available products -- MY UV PATCH, UV Tester, etc -- that can operate with or without electronic components. The authors capabilities in UVC sensing are distinct, but of no practical consequence. As a result, I view the findings to be incremental. In any case, the suitability for publication represents a judgement best left to the editors.

Reviewer #3 (Remarks to the Author):

The responses have been edited for clarity and brevity. Now the reviewer agree this revised manuscript is published

We thank the reviewers for their time and critical assessment of our work throughout the review process. This has assisted in improving the quality of this work. A response to the reviewers' comments is provided below:

Response to Reviewer #1

Comment: The authors provide some useful responses to reviewer comments, but my original judgement that this manuscript does not rise to a level required for a publication in Nature Communication remains. The basic concept of using color change chemistries to monitor UV exposure in a thin flexible device are already in the literature (for UVA and for UVA and UVB separately), and they form the basis for several commercially available products -- MY UV PATCH, UV Tester, etc -- that can operate with or without electronic components. The authors capabilities in UVC sensing are distinct, but of no practical consequence. As a result, I view the findings to be incremental. In any case, the suitability for publication represents a judgement best left to the editors.

Response: Thank you for your opinion. We appreciate that the reviewer identifies the distinct ability of our sensor to differentiate UVC from UVA and B. At the same time, we respectfully disagree that UVC sensing does not have practical consequences. As highlighted in our previous response to reviewers, the rapid increase in the use of UVC as disinfectant lamps in the household and consumer settings will create a niche need for such sensors. Similarly, we disagree that any of the UV-induced colour change chemistry reported in the literature follows the same scientific rationale as proposed in the current work. Our current work has for the first time shown the importance of multi-redox photo-electrochromic molecules in spectral-differentiation of different wavelengths of light. We appreciate the Editor's judgement to support this work.

Response to Reviewer #3

Comment: The responses have been edited for clarity and brevity. Now the reviewer agree this revised manuscript is published

Response: Thank you.